# Oligo-Fucoidan Prevents M2 Macrophage Differentiation and HCT116 Tumor Progression

**DOI:** 10.3390/cancers12020421

**Published:** 2020-02-12

**Authors:** Li-Mei Chen, Hong-Yu Tseng, Yen-An Chen, Aushia Tanzih Al Haq, Pai-An Hwang, Hsin-Ling Hsu

**Affiliations:** 1Institute of Molecular and Genomic Medicine, National Health Research Institutes, Miaoli 35053, Taiwan; limei@nhri.org.tw (L.-M.C.); hytseng@nhri.org.tw (H.-Y.T.); neuropt438@nhri.org.tw (Y.-A.C.); aushia.tanzia@nhri.edu.tw (A.T.A.H.); 2National Taiwan Ocean University, Keelung 20224, Taiwan; amperehwang@gmail.com

**Keywords:** Oligo-Fucoidan, antioxidant, cisplatin, ROS, M1/M2 macrophages, tumor progression

## Abstract

Reactive oxygen species (ROS) produced during intracellular metabolism or triggered by extrinsic factors can promote neoplastic transformation and malignant microenvironment that mediate tumor development. Oligo-Fucoidan is a sulfated polysaccharide isolated from the brown seaweed. Using human THP-1 monocytes and murine Raw264.7 macrophages as well as human HCT116 colorectal cancer cells, primary C6P2-L1 colorectal cancer cells and human MDA-MB231 breast cancer cells, we investigated the effect of Oligo-Fucoidan on inhibiting M2 macrophage differentiation and its therapeutic potential as a supplement in chemotherapy and tumor prevention. We now demonstrate that Oligo-Fucoidan is an antioxidant that suppresses intracellular ROS and mitochondrial superoxide levels in monocytes/macrophages and in aggressive cancer cells. Comparable to ROS inhibitors (DPI and NAC), Oligo-Fucoidan directly induced monocyte polarization toward M1-like macrophages and repolarized M2 macrophages into M1 phenotypes. DPI and Oligo-Fucoidan also cooperatively prevented M2 macrophage invasiveness. Indirectly, M1 polarity was advanced particularly when DPI suppressed ROS generation and supplemented with Oligo-Fucoidan in the cancer cells. Moreover, cisplatin chemoagent polarized monocytes and M0 macrophages toward M2-like phenotypes and Oligo-Fucoidan supplementation reduced these side effects. Furthermore, Oligo-Fucoidan promoted cytotoxicity of cisplatin and antagonized cisplatin effect on cancer cells to prevent M2 macrophage differentiation. More importantly, Oligo-Fucoidan inhibited tumor progression and M2 macrophage infiltration in tumor microenvironment, thus increasing of anti-tumor immunity.

## 1. Introduction

The malignant tumor microenvironment (TME) governs tumor progression [1], metastasis and recurrence. The cytokines, chemokines and/or growth factors produced by cancerous and stromal cells can cause tumor aggressiveness and/or therapeutic resistance that determine the overall and recurrence-free survival rates of patients [2,3,4,5]. Tumor-associated macrophages (TAMs) undergo M1 or M2 macrophage polarization in response to different stimuli [6]. M1 macrophages are activated by lipopolysaccharide (LPS) and IFN-γ produced by Th1 cells, while M2 macrophages are activated by interleukin (IL)-4 and IL-13 produced by Th2 cells. Classically activated M1 macrophages exhibit pro-inflammatory phenotypes and produce ROS, nitric oxide (NO) and proinflammatory cytokines (such as TNF-α, IL-1, IL-6, IL-12 and IL-23) [7,8]. Alternatively activated M2 macrophages regulate anti-inflammation and reduce NO and reactive oxygen species (ROS) levels but induce Arginase-1, anti-inflammatory cytokines (such as TGF-β, IL-10 and IL-6) and growth factors that support neoplasia [8,9] and induce tumor cell movement [10].

Redox metabolism regulates macrophage polarization. Superoxide (O_2_^−^) generated by either cytoplasmic membrane-bound nicotinamide-adenine dinucleotide phosphate (NADPH) oxidase or the mitochondrial electron transfer chain is converted into H_2_O_2_ by superoxide dismutase (SOD) [11], which can stimulate M2 polarization via STAT6 signaling activation. High endoplasmic reticulum (ER) stress also switches macrophage polarity from the M1 phenotype into the M2 phenotype [12]. However, the molecular mechanism of macrophage polarization underlying the metabolic programming is still ambiguous; in particular, the impact of ROS on macrophage differentiation is controversial, possibly due to the inducers and tissue-specificity of ROS and the status of oxidases or ROS scavengers. Thereby, ROS modulators may redirect macrophage plasticity and improve disease therapy.

Seaweed polysaccharides have been recommended as a supplement for health enhancement and disease management. Oligo-Fucoidan (also known as Low-Molecular-Weight Fucoidan; LMF) is a sulfated polysaccharide purified from *Sargassum hemiphyllum* [13]. Oligo-Fucoidan can attenuate the negative effects of etoposide (ETO) chemotherapy that promotes IL-6 and MCP-1/CCL2 production in aggressive HCT116 cancer cells which activate Stat3 and Stat6 signaling [13], respectively. ETO treatment of cancer cells activates the IL-6/JAK1/STAT3 pathway that may mediates communication between tumor cells and the microenvironment [14,15,16]. Therefore, Oligo-Fucoidan supplementation may prevent IL-6/CCL2-mediated epithelial-mesenchymal transition and M2-type macrophage polarization as previously described [17,18]. Besides, Oligo-Fucoidan supplementation potentially represses tumor angiogenesis and metastasis as well as the negative effects of gemcitabine and cisplatin chemotherapy [19,20,21,22], which induce cancer cachexia-related muscle atrophy in bladder cancer-bearing mice. Our previous studies have proven that Oligo-Fucoidan enhances the function of p53 and the checkpoint control of the G2/M phase upon ETO treatment [13], these may impede tumorigenicity [23], cancer stemness and tumor relapse. Besides, Oligo-Fucoidan supplementation sensitizes lung cancer cell cytotoxicity induced by cisplatin via stimulation of TLR4/CHOP-mediated caspase-3 and PARP activity that prevent tumor development in mice [24]. More importantly, the clinical trials have indicated that patients with lung cancer co-administered with cisplatin and Oligo-Fucoidan showed better clinical outcomes.

We now confirm that Oligo-Fucoidan quenches intracellular ROS and mitochondrial superoxide production that benefit M1-like macrophage polarization from monocytes and M0 macrophages. Oligo-Fucoidan supplementation sufficiently advances cytotoxicity of cisplatin in aggressive cancer cells and indirectly attenuates drawback of cisplatin on M2 macrophage promotion, by which renovates the microenvironment to prevent tumor progression.

## 2. Results

### 2.1. ROS Inhibitors and Oligo-Fucoidan Advance M1-Like Macrophage Polarization

ROS can induce TAM recruitment and M2 polarization [25]. Mitochondria are central to superoxide production [26]. The active phosphorylation of p47^phox^, a major regulatory subunit of NADPH oxidase, stimulates intracellular superoxide generation via the redox signaling pathway [27,28]. Diphenyleneiodonium (DPI), an inhibitor of nicotinamide-adenine dinucleotide phosphate (NADPH) oxidase in mitochondrial electron transport chain Complex I [29], suppresses mitochondrial ROS generation. When the intracellular ROS levels were analyzed in unprimed THP-1 monocytes upon DPI treatment (Figure 1A), DPI decreased ROS production in a dose-dependent manner, as shown by reduced oxidization of 2’,7’-dichlorofluorescin diacetate (DCF-DA) into fluorescent 2’,7’-dichlorofluorescein (DCF). While DPI prevented the phosphorylation of p47^phox^ (Ser345) (p-p47^phox^) in monocytes (Figure 1B), the levels of intracellular M1 marker, iNOS and the phosphorylated p38 (Thr180/Tyr182) (p-p38), were also increased. Macrophage polarity detected by flow cytometry revealed that the populations of CD80(+) M1 macrophages were also expanded upon DPI treatment (Figure 1C) compared with MOCK and isotope IgG control (Appendix A), suggesting that ROS inhibition favored M1-like plasticity. 

Comparing with antioxidants, DPI and N-acetylcysteine (NAC), we identified that Oligo-Fucoidan also decreased THP-1 cellular ROS (Figure 1D), as indicated by reduced oxidation of DCF-DA. In response to DPI, NAC and Oligo-Fucoidan treatment, the treated THP-1 monocytes showed the induced M1-like macrophage marker (CD68 and CD80) (Figure 1E) and the reduced M2 macrophage marker TGF-β (Figure 1F). Consistently, these antioxidants increased the M1 marker p-p38 (Thr180/Tyr182) and suppressed the M2 marker CD163 (Figure 1G).

To activate M0, M1 and M2 macrophage differentiation, THP-1 monocytes were treated with PMA to induce M0 macrophage polarization, followed by LPS or IL-4 stimulation, respectively. The macrophage phenotypes were confirmed by analyzing the specific marker expression with quantitative RT-PCR. F4/80 mRNA level was increased in the M0 and M1 macrophages (Appendix A), CD80 and CD86 mRNA levels were more induced in the M1 macrophages (Appendix A) but CD163 and CD206 mRNA levels were promoted in only the M2 macrophages (Appendix A). Correspondingly, F4/80 protein was more elevated in all M0, M1 and M2 macrophages than monocytes (Appendix A). However, iNOS, p-p38 (Thr180/Tyr182), CD80 and TNF-α proteins were much enriched in the M1 macrophages than those of CD163, Arginase-1 and IL-10 proteins specifically increased in the M2 macrophages. 

By evaluating M2 macrophages’ response, we also found that the expression levels of M1 marker CD68 and CD86 were much induced upon DPI and Oligo-Fucoidan treatment of the differentiated THP-1 M2 macrophages than NAC (Figure 1H) and the percentages of CD163(+)/CD206(+) M2 macrophages were significantly reduced (Figure 1I), as compared with untreated group (MOCK) and isotope IgG control (Appendix A); revealing that antioxidants repolarized M2 macrophages.

Furthermore, we compared the antioxidant potential of Oligo-Fucoidan (LMF) (0.5–0.8 kDa) from *Sargassum hemiphyllum* and high molecular weight fucoidan (HMF) (20–200 kDa) from *Fucus vesiculosus*. As the reduced oxidation MitoSOX Red in the treated THP-1 monocytes, both LMF and HMF suppressed mitochondrial superoxide levels (Appendix A). Correspondingly, the phosphorylation of NADPH oxidase subunit p47^phox^ on Ser345 was dramatically reduced upon LMF or HMF treatment (Appendix A). Intriguingly, the THP-1 monocyte-differentiated M2 macrophages treated by LMF or HMF were also suppressed the populations of CD163(+) and CD206(+) M2 macrophages (Appendix A), as compared with MOCK and isotype IgG control (Appendix A).

Upon DPI treatment, THP-1 monocyte-differentiated macrophages were induced mRNA expression of marker of M0 (F4/80) macrophages (Appendix A) and M1 (CD86) macrophages (Appendix A), which were advanced by simultaneous Oligo-Fucoidan and DPI treatment, signifying a cooperative effect on M1 promotion. When the THP-1-differentiated M2 macrophages were administrated with DPI and/or Oligo-Fucoidan, the marker of M0 (F4/80) (Appendix A) and M1 (CD68, CD80, CD86) (Appendix A) macrophages were all significantly promoted.

These results indicate that antioxidants and Oligo-Fucoidan together advance monocyte polarization toward M1 polarity and further repolarize M2 macrophages into M1 plasticity.

### 2.2. ROS Produced in Cancer Cells Influence Macrophage Differentiation

To inspect whether ROS produced in cancer cells can impact macrophage plasticity, two isogenic HCT116 colon cancer cell lines without (p53^−/−^) or with (p53^+/+^) wild-type p53 were treated with DPI. When DPI suppressed intracellular ROS (Figure 2A) and mitochondrial superoxide levels (Figure 2B), the phosphorylation of p47^phox^ (Ser345) was significantly decreased by DPI independent of the p53 status (Figure 2C) and DPI dramatically induced the active phosphorylation of p53 (Ser15) and the accumulation of p53 in the p53^+/+^ cancer cells. 

Next, HCT116 cells were treated with DPI (1 M) and/or Oligo-Fucoidan (400 g/mL) and the preconditioned cancer cells were placed in the upper compartment of a Boyden chamber transwell and cocultured with THP-1 monocytes placed in the bottom chamber for 48 h. The results showed that the DPI-pretreated cancer cells stimulated monocyte polarization toward the phenotypes of F4/80 (high) M0 (Figure 2D) and CD80 (high) M1 (Figure 2E) macrophages. Interestingly, the Oligo-Fucoidan-treated cancer cells also upregulated F4/80 and CD80 mRNA levels (Figure 2D,E) and these effects were additively enhanced by DPI cotreatment. Thus, ROS inhibition in the cancer cells can indirectly promote M0 and M1 differentiation.

Although M2 macrophages were highly invasive (Figure 2F), combined DPI and Oligo-Fucoidan treatment significantly suppressed their invasive abilities than the individual treatments. Reciprocally, the invasiveness of HCT116 cancer cells was markedly inhibited when they encountered the M2 macrophages those were pretreated by DPI and/or Oligo-Fucoidan (Figure 2G). Accordingly, DPI and Oligo-Fucoidan treatment can suppress M2 macrophage invasion and renovate M0 and M1 plasticity which prevent cancer cell invasion. 

### 2.3. Cisplatin and Oligo-Fucoidan Cooperatively Promote the Polarization of M1 Macrophages

To further analyze the chemotherapeutic effect on macrophage plasticity, THP-1 monocytes were treated with cisplatin (15 μM) and/or Oligo-Fucoidan (400 μg/mL). As detected by WST-1 cell viability assay, THP-1 cell viability was significantly reduced (41%) by cisplatin but less affected by Oligo-Fucoidan (87%) compared with MOCK treatment (100%) (Appendix A). Mitochondrial superoxide level of THP-1 cells was induced by cisplatin (1.35-fold) but suppressed by Oligo-Fucoidan (0.75-fold) and by combined treatment (0.78-fold) relative to MOCK treatment (1-fold) (Figure 3A). Analysis of macrophage polarity by flow cytometry indicated that cisplatin expanded the CD80(+) M1 populations (57.7%) and that was further increased by Oligo-Fucoidan supplementation (69.2%) (Figure 3B), comparing with MOCK (0.6%) and isotope IgG controls (Appendix A). Unexpectedly, cisplatin also induced the CD163(+) M2 population size (51.1%) (Figure 3C) but this effect was decreased by adding Oligo-Fucoidan (34.9%). Similar to M0 marker F4/80 mRNA levels were upregulated by cisplatin (2.61-fold) and additively enhanced by Oligo-Fucoidan supplement (3.70-fold) (Figure 3D), M1 marker CD68 mRNA levels were induced by cisplatin (3.75-fold) and Oligo-Fucoidan (2.74-fold) as well as in combined treatment (4.37-fold) (Figure 3E). Cisplatin treatment also enhanced the M2 phenotypes with increased CD206 (24.89-fold) mRNA expression in THP-1 cells (Figure 3F) but Oligo-Fucoidan supplementation antagonized this effect and reduced CD206 (9.43-fold) mRNA level.

Likewise, M0 macrophages derived from PMA-treated THP-1 monocytes were polarized into CD86(+) M1 populations especially in combined treatment with cisplatin and Oligo-Fucoidan compared with individual treatments (Figure 3G), MOCK and isotope IgG control (Appendix A). Cisplatin also mediated M0 macrophages polarizing into CD206(+) M2 populations (52.5%) (Figure 3H) and Oligo-Fucoidan supplementation reduced this side effect (30.8%).

Raw264.7 cells are monocytes/macrophages derived from BALB/c mice. Interestingly, Raw264.7 macrophages were also found to more increase CD80(+)/CD86(+) M1 populations upon cisplatin (50.3%), Oligo-Fucoidan (46.3%) and combination (59.2%) treatment than MOCK (34%) (Figure 3I) and isotype IgG control (Appendix A). Similarly, Raw264.7 macrophages highly expressed M1 marker (CD68 and CD80) upon combined treatment than individual effects (Appendix A) and Oligo-Fucoidan supplementation reduced the side effect of cisplatin on stimulating M2 marker (CD206 and Arginase-1 (Arg-1)) expression (Appendix A).

Subsequently, Raw264.7 macrophages were treated with IL-4 (20 ng/mL) for 48 h to induce M2 macrophage polarization. While administrating the M2 macrophages with cisplatin and/or Oligo-Fucoidan), CD80(+) M1 populations were much improved (Appendix A) and M1 marker (CD68 and CD80) expression levels were promoted particularly in combined treatment (Appendix A). Though cisplatin treatment still enriched M2 marker (CD206 and Arg-1) expression, Oligo-Fucoidan supplementation blunted this negative effect (Appendix A). Hence, Oligo-Fucoidan may improve the chemotherapeutics via M1 enrichment and M2 inhibition.

### 2.4. Cisplatin and Oligo-Fucoidan Additively Enhance Cancer Cell Death and Polarization of M1 Macrophages 

The p53-p21 pathway induces Cyclin B1 degradation which reinforces G2/M arrest and cell death [30,31,32]. Cisplatin (15 μM) treatment induced the accumulation and phosphorylation (Ser15) of p53 and activated the phosphorylation (Ser139) of histone H2AX (γ-H2AX) (a biomarker of DNA double-strand breaks) in the p53^+/+^ cells more strongly than in the p53^−/−^cells (Appendix A). Differently, Oligo-Fucoidan (400 μg/mL) treatment effectively suppressed intrinsic DNA lesions, thus decreasing γ-H2AX level (lanes 3 and 7) than that seen with MOCK treatment (lanes 1 and 5). While cisplatin and Oligo-Fucoidan cooperatively enhanced p53-mediated p21 induction and Cyclin B1 reduction in the p53^+/+^ cells (Appendix A, lane 8), Cyclin B1 was constitutively expressed in the p53^−/−^ cells (lanes 1–4).

Although Oligo-Fucoidan treatment did not affect the distributions of the p53^−/−^ cancer cells and the p53^+/+^ cancer cells in the cell cycle (Appendix A), cisplatin induced sub-G1 populations, a sign of apoptosis with reduced DNA content, more in the p53^+/+^ cells (57.5%) (Appendix A) than in the p53^−/−^ cells (9.2%) (Appendix A). However, cisplatin had a stronger G2/M phase-arresting effect on the p53^−/−^ cells (52.8%) than on the p53^+/+^ cells (17.2%). Combined treatment also increased more G2/M arrest in the p53^+/+^ cells (25.1%) than cisplatin monotherapy (17.2%) (Appendix A). Histograms display the p53^−/−^ and p53^+/+^ cell cycle distributions for the indicated treatment. Likewise, we have found that Oligo-Fucoidan and ETO collaboratively enhanced p53 function and G2/M arrest alongside inhibition of the CDC2/Cyclin B1 pathway [13]. Hence, Oligo-Fucoidan supplementation may improve the chemotherapeutics, especially in the presence of p53.

Using MitoSOX Red oxidation and flow cytometry analysis, we observed that cisplatin greatly provoked mitochondrial ROS production in the p53^−/−^ cells (92.5%) (Figure 4A) and the p53^+/+^ cells (96.1%) (Figure 4B) compared with MOCK controls (54.5% in the p53^−/−^ cells and 65.2% in the p53^+/+^cells), while Oligo-Fucoidan significantly decreased mitochondrial superoxide levels in the p53^−/−^ cells (24.9%) (Figure 4A) and the p53^+/+^ cells (39.6%). Though cisplatin promoted mitochondrial ROS in the p53^−/−^ cells (1.7-fold) (Figure 4A) and the p53^+/+^ cells (1.5-fold) (Figure 4B), Oligo-Fucoidan supplement reduced mitochondrial ROS in the p53^−/−^ cells (1.3-fold) but not the p53^+/+^ cells (1.5-fold). 

Manganese superoxide dismutase (MnSOD) [33], an O^2^^−^ scavenger in mitochondria, is overexpressed in carcinogenesis. Glutathione peroxidase (GPX) diminishes ROS buildup and reduces oxidative damage [34]. Under the oncogenic stress [35], YY1 induction and EGFR activation are stimulated that lead to MnSOD expression and ROS generation. Similarly, cisplatin and Oligo-Fucoidan collectively amplified the ROS signaling pathway in HCT116 (p53^−/−^ and p53^+/+^) cells (Appendix A), which showed increase of YY1, EGFR, EGFR phosphorylation (p-EGFR) (Tyr1068), MnSOD and γ-H2AX but decrease of GPX. To confirm whether combined treatment indeed amplifies the oxidative damaging cascade, ROS inhibitors (NAC, MitoQ and DPI) were treated HCT116 cancer cells followed by administration with cisplatin and Oligo-Fucoidan (Appendix A). The results showed that YY1, p-EGFR (Tyr1068), ROS scavenger (MnSOD and catalase) and the phosphorylation (Ser15) and accumulation of p53 were particularly reduced by MitoQ and DPI, representing that combined treatment indeed promoted the ROS signaling.

Excessive mitochondrial ROS levels cause adenosine triphosphate (ATP) exhaustion and cell death [36]. An examination of cancer cell death in response to cisplatin and/or Oligo-Fucoidan confirmed that combined treatment further promoted the cleaved PARP (Asp214) and active caspase 3 levels compared with monotherapies (Figure 4C, lane eight vs. lanes six and seven), particularly in the p53^+/+^ context. Moreover, FITC-Annexin V and PI staining, followed by flow cytometry analysis revealed that cisplatin induced more necrotic and late apoptotic events in p53^+/+^ cells (13% and 66.4%, respectively) (Figure 4E) than in p53^−/^^−^ cells (3.1% and 20%, respectively) (Figure 4D). Combined treatment advanced the outcomes of p53^−/^^−^ cell apoptosis (34.3%) (Figure 4D) and p53^+/+^ cell necrosis (39.1%) (Figure 4E), thus exacerbating the cancer cell death. Histograms indicate the necrotic, early and late apoptotic results in individual treatments of the p53^−/^^−^ cells and the p53^+/+^ cells. 

In study of the primary C6P2-L1 cell lines derived from the colorectal cancer patients (Figure 5), we also identified that Oligo-Fucoidan supplementation advanced cisplatin effect on inducing the PARP (Asp214) cleavage and caspase 3 activation (Figure 5A), thus evolving the apoptotic events more than the individual treatments (Figure 5B). Clearly, although Oligo-Fucoidan has antioxidative ability, it would not defeat cytotoxic effect of cisplatin in treatment of cancer cells.

To inspect whether the cisplatin and/or Oligo-Fucoidan-treated cancer cells can influence macrophage plasticity, THP-1 monocytes were incubated with the pre-conditioned HCT116 cells in a Boyden chamber transwell. The results indicated the THP-1 cells increased F4/80 (M0 marker) mRNA levels after incubation with the p53^−/^^−^ cells (3.28-fold) or the p53^+/+^ cells (2.24-fold) those were experienced combination treatment more than monotherapies (Figure 6A). In addition, the enriched CD86 mRNA levels were detected when THP-1 monocytes encountered the p53^−/^^−^ cells or the p53^+/+^ cells pre-treated with cisplatin or Oligo-Fucoidan (Figure 6B); in particular, M1 polarity was advanced by the p53^−/^^−^ cells (2.35-fold) or the p53^+/+^ cells (2.11-fold) those were cotreated by both agents. A flow cytometry study also confirmed that CD80(+) M1 populations were expanded when monocytes encountered the p53^−/^^−^ cells (Appendix A) or the p53^+/+^ cells (Appendix A) experienced combined treatment.

Unexpectedly, the cisplatin-treated p53^−/^^−^ cancer cells also activated monocyte polarization into the M2 phenotype with CD206 increase (1.16-fold) while the Oligo-Fucoidan-treated p53^−/^^−^ cells decreased the CD206 level (0.63-fold) compared to MOCK control (Figure 6C). However, Oligo-Fucoidan supplementation antagonized the negative effect of cisplatin on the p53^−/^^−^ cells (Figure 6C), suppressing the CD206 level to 0.65-fold in the polarized macrophages. Similar results were identified in the p53^+/+^ cancer cells experienced cisplatin and/or Oligo-Fucoidan treatment (Figure 6C). Consistently, the polarized THP-1 cells revealed the induced amounts of M1 marker (iNOS and CD80) and the reduced amounts of M2 marker (CD163 and Arginase-1) while reacting with the cancer cells those were pretreated with Oligo-Fucoidan or in combination with cisplatin (Figure 6D). Also, THP-1 cells encountered breast cancer MDA-MB 231 cells those were pretreated with cisplatin and/or Oligo-Fucoidan expressed the higher mRNA levels of M0 (F4/80) (Appendix A) and M1 (CD68 and CD86) (Appendix A) marker. Moreover, Oligo-Fucoidan supplementation repressed the mRNA levels of M2 marker (CD163 and IL-1β) induced by cisplatin treatment (Appendix A).

To analyze M2 macrophage invasiveness in the treatment, THP-1 monocytes were stimulated with PMA and then IL-4, the M2 macrophage invasion ability was substantially suppressed upon cisplatin and/or Oligo-Fucoidan treatment (Figure 6E). The M2 macrophage polarity also significantly shifted toward the M0 and M1 phenotypes in combined treatment, as indicated by increasing of F4/80 (Figure 6F) and CD80 (Figure 6G) but decreasing of CD163 (Figure 6H) mRNA levels.

Therefore, Oligo-Fucoidan and/or cisplatin can directly impede M2 macrophage invasion and repolarize the M2 phenotype to the M0 and M1 phenotypes. Further, the cancer cells pre-treated with Oligo-Fucoidan alone or in combination with cisplatin can indirectly promote M0 and M1 plasticity and suppress M2 polarity. 

### 2.5. Oligo-Fucoidan Inhibits Tumor Progression and M2 Macrophage Infiltration

To investigate the therapeutic effect of Oligo-Fucoidan and/or cisplatin, HCT116 cells (2 × 10^6^) were subcutaneously injected into BALB/c nude mice. Two weeks after inoculation, the xenograft mice were treated with cisplatin for 2 weeks and/or supplemented with Oligo-Fucoidan for 5 weeks (Figure 7A). Control mice were processed with phosphate-buffered saline (PBS) alone. Tumor growth rates were suppressed dramatically in the p53^+/+^ cancer cell-bearing mice upon Oligo-Fucoidan (F) or in combination with cisplatin treatment (F + C) compared with cisplatin (C) or PBS treatment (Figure 7B). However, neither the monotherapy (C or F) nor combined therapy (C + F) significantly affected p53^−/^^−^ tumor development (Figure 7C). At week 7, the p53^−/^^−^ tumor burdens were more progressive than the p53^+/+^ tumor burdens (Figure 7D). Although cisplatin alone (C) did not effectively decrease p53^−/^^−^ tumor progression, Oligo-Fucoidan alone (F) and in combination with cisplatin (C + F) improved the therapeutic efficacy. Importantly, p53^+/+^ tumor regression was achieved in mice upon Oligo-Fucoidan monotherapy (F) or combination therapy (C + F) (Figure 7E), which also inhibited the tumor necrosis (denoted by arrowheads). As indicated by immunohistochemistry (IHC) staining of vascular endothelial growth factor receptor 2 (VEGFR2) (Figure 7F), Oligo-Fucoidan (F) alone or combined treatment (C + F) also repressed the angiogenesis effect in the p53^+/+^ and p53^−/^^−^ tumors, while cisplatin only well inhibited the p53^+/+^ tumor angiogenesis. Hence, Oligo-Fucoidan alone or combined with cisplatin substantially impedes tumor progression and angiogenesis, especially in presence of p53.

Afterward, TAMs were assessed by IHC analysis, showing that more CD163(+) M2 macrophages (indicated by asterisks) infiltrated into the PBS-treated p53^−/^^−^ tumors than the Oligo-Fucoidan- or cisplatin-treated p53^−/^^−^ tumors (Figure 8A). Also, more M2 macrophages were accumulated in the stromal region of the PBS-treated p53^−/^^−^ tumors than that of the Oligo-Fucoidan- or cisplatin-treated p53^−/^^−^ tumors (Figure 8B) and the retention of M2 macrophages in tumoral and stromal regions were greatly inhibited by combined therapy (C + F) (Figure 8A,B).

Also, M2 macrophage infiltration was more abundant in the PBS-treated p53^+/+^ tumors and they were noticeably reduced by cisplatin and/or Oligo-Fucoidan treatment (Figure 8C). As compared with the p53^−/−^ tumors (Figure 8B), the p53^+/+^ tumors had less stromal M2 macrophage accumulation (Figure 8D), which were also reduced by each monotherapy or eliminated by combo therapy. IHC images (2× magnification) of the tumor and stroma regions in the indicated treatment were shown in Appendix A. Taking together, Oligo-Fucoidan alone or in combination with cisplatin capably prevents M2 macrophage infiltration that renovates the tumor suppressive microenvironment.

## 3. Discussion

The oxidative microenvironment fuels the function and recruitment of M2 macrophages and TAMs [25] and the high mitochondrial ROS can induce cell malignancy and tumor initiation [37]. Overexpression of Cu-SOD and Zn-SOD catalyze the generation of H_2_O_2_ which can serve as the redox switch to induce polarization of M2 macrophages [11], enhancing Arginase-1 and urea levels but reducing the iNOS levels and NO synthesis required for M1 differentiation. ROS accumulation in cancerous or stromal cells may activate inflammatory factors [38,39,40], which promote cell transformation and tumor initiation, progression and metastasis. 

H_2_O_2_ oxidant also modulates the activity of YY1 [41] and EGFR [42] by oxidative modification of cysteine residues. Paradoxically, the oxidative stress-provoked YY1 expression potentiates antioxidant machinery in irradiation-induced neuronal damage via amplification of NRF2-mediated transcriptional activation of antioxidant responsive elements [43], thereby protecting neuronal cells against oxidative damage. Conversely, NRF2 promotes EGFR expression to fuel mRNA translation in maintenance of pancreatic tumor [44]. Furthermore, K-Ras(G12D) [45], B-Raf(V619E) and Myc oncogenes induce NRF2 antioxidant program, by which enhance ROS detoxification and tumorigenesis. Besides, the activation of MCT-1 oncogene capably induces YY1-EGFR signaling axis that promotes MnSOD expression [35], mitochondrial ROS generation, lung cancer cell invasion and tumor progression. Here, we identify that Oligo-Fucoidan combined with cisplatin treatment induces ROS signaling in colorectal cancer cells which can be suppressed by antioxidants (NAC, MitoQ and DPI) (Appendix A), demonstrating that Oligo-Fucoidan and cisplatin together induce the oxidative signaling axis of YY1/EGFR/MnSOD (Appendix A). How this antioxidant mechanism in cancer cells reprogram the macrophage polarity remained unclear.

We found that Oligo-Fucoidan is an idea antioxidant that decreases mitochondrial superoxide and intracellular ROS in monocytes (Figure 1D and Figure 3A) and cancer cells (Figure 4A,B). Importantly, M1-like polarity was promoted after inhibiting intracellular ROS generation in monocytes (Figure 1A–E), supporting that an antioxidative mechanism can directly activate tumor suppressive function of macrophages (Figure 9). Furthermore, DPI and Oligo-Fucoidan cotreatment directly activated polarization of M0 and M1 from monocytes (Appendix A) and M2 macrophages (Appendix A) and DPI and Oligo-Fucoidan synergistically prohibited M2 macrophage invasiveness (Figure 2F). Moreover, LMF (Oligo-Fucoidan) or HMF treatment of M2 macrophages decreased the populations of CD163(+) and CD206(+) M2 macrophage (Appendix A). 

When M2 macrophages were treated with DPI and/or Oligo-Fucoidan (Figure 2F), the invasiveness of M2 macrophages was effectively inhibited. Mutually, the cancer cell invasion ability enhanced by the M2 macrophages was repressed while pre-treating the M2 macrophages with DPI and/or Oligo-Fucoidan (Figure 2G). 

As cisplatin chemotherapy produced high levels of mitochondrial superoxide in monocytes (Figure 3A), monocytes were differentiated into not only M0 (Figure 3D) and M1 (Figure 3B,E) but also M2 (Figure 3C,F) phenotypes. Similarly, cisplatin activated the M0 macrophage polarization into not only M1 (Figure 3G) but also M2 (Figure 3H) types, whereas Oligo-Fucoidan supplementation blunted cisplatin effect on promoting M2 polarity. Thus, Oligo-Fucoidan supplement may capably relief the problem of M2 promotion after chemotherapy. ROS inhibitors such as DPI, NAC, MitoQ and Oligo-Fucoidan may proficiently prevent M2 macrophage differentiation and attenuate the chemotherapeutic side effect and renovate a healthy microenvironment. It will be important to inspect whether antioxidant(s) can reduce M2 macrophage infiltration in tumors.

Fucoidan also regulates immune responses that affect the production of chemokines and proinflammatory cytokines as well as the number and activity of immune cells [46,47,48,49]. For example, the expression of the M2-type chemokine CCL22 through the NF-κB pathway in M2 macrophages is downregulated by Fucoidan [50], which may inhibit tumor cell migration and regulatory T cell recruitment. These suggest that Fucoidan supplementation may alter cancer immunity that helps disease treatment. We have demonstrated that Oligo-Fucoidan reduces IL-6 and MCP-1/CCL2 expression and secretion in the ETO-treated cancer cells [13]; thus, Oligo-Fucoidan may control autocrine loops in cancer cells and paracrine pathways in the innate and/or adaptive immune systems. By changing the oxidative metabolism and cytokine/chemokine profile of the immune cells and chemotherapeutic cells, different types of Fucoidan may improve therapeutics and disease-free survival and attenuate chemotherapy-related illnesses, distress and/or inflammation in patients. 

Although Oligo-Fucoidan reduces ROS generation, it would not weaken the oxidative damage and cytotoxicity in the chemotherapeutic cancer cells (Figure 4 and Figure 5). In combination with cisplatin, Oligo-Fucoidan still promoted cancer cell death through increasing p53-p21 signaling (Appendix A), PARP cleavage and caspase-3 activation (Figure 4C and Figure 5A), which may cause the cytotoxicity. Oligo-Fucoidan supplementation not only advanced cisplatin’s cytotoxic effects on colon cancer cells (Figure 4D,E) and breast cancer cells (Figure 5), Oligo-Fucoidan also promoted TGF-β receptor degradation and Toll-like receptor 4-mediated pathway to induce ER stress [51,52], which triggered lung cancer cell death and executed a tumor-suppressing mechanism [51]. 

Algae species diversity, growing conditions and purification processes could affect the structure and activity of Fucoidan as well as the therapeutic efficacy. It has been shown that fucoidans mainly act via the PI3K/AKT [53], MAPK and caspase signaling pathways. Although the structure of Oligo-Fucoidan is still unresolved, its functions are similar to that of HMFs in anti-lung cancer [51], genomic protection [13] (Appendix A, lanes 3 and 7) and the M2 inhibition [50]; thus, they may share analogous molecular structures as described previously [54,55,56,57].

Consistent with cisplatin’s direct effect on monocytes and M0 macrophages (Figure 3), the cisplatin-treated cancer cells not only stimulated monocytes differentiating into M0 (Figure 6A) and M1 (Figure 6B and Appendix A) but also M2 phenotypes (Figure 6C); however, the M2 polarization promoted by the cisplatin-treated cancer cells was repressed by Oligo-Fucoidan supplement. In particular, while monocytes encountered the p53^+/+^ cancer cells pretreated with Oligo-Fucoidan and cisplatin (Figure 6D), the amounts of M1 marker (iNOS and CD80) were relatively increased but those of M2 marker (CD163 and Arginase-1) were significantly decreased. Combined treatment also directly inhibited the invasiveness of M2 macrophages (Figure 6E), reduced the M2 phenotype (Figure 6H) and repolarized the M2 macrophages toward the M0 and M1 phenotypes (Figure 6F,G). Consequently, the development and progression of p53^+/+^ tumors were more inhibited by Oligo-Fucoidan alone and in combination with cisplatin (Figure 7B,D,E), as compared with the p53^−/−^ tumors (Figure 7C–E). Similarly, M2 macrophage recruitment in the p53^−/−^ tumoral and stromal regions were more abundant (Figure 8A,B) than in the p53^+/+^ microenvironment (Figure 8C,D). Oligo-Fucoidan monotherapy or combined with cisplatin successfully inhibited M2 macrophage infiltration (Figure 8), signifying that the suppressive TME were renovated. Further inspection of the mechanism(s) of Oligo-Fucoidan and its systematic activity with current drugs or newly developed immunotherapies in anti-tumor immune surveillance will be an important subject.

## 4. Materials and Methods

### 4.1. Cell Lines

Human colorectal carcinoma HCT116 cell lines with or without wild-type p53 expression (p53^+/+^ and p53^−/−^), human invasive breast cancer MDA-MB-231 cells, human THP-1 monocytes and murine macrophage-like RAW264.7 cells were cultured in RPMI-1640 medium (Thermo Fisher Scientific, Waltham, MA, USA) supplemented with 10% fetal bovine serum (FBS) (Thermo Fisher Scientific), L-glutamine (2 mM) (Thermo Fisher Scientific), penicillin (100 units/mL) and streptomycin (100 μg/mL) (Thermo Fisher Scientific) in a 37 °C incubator with 5% CO_2_ and 95% humidity. 

Colorectal cancer from the patient was minced and dissociated with 200 U/mL collagenase type IV (Sigma-Aldrich, St. Louis, MO, USA) and passed through a Falcon 40 M cell strainer (Corning, Corning, NY, USA). Red blood cells were lysed with ammonium–chloride-potassium buffer (Thermo Fisher Scientific). The isolated primary cancer cell line (C6P2-L1) was first grown in Dulbecco’s Modified Eagle Medium (DMEM) containing 20% FBS, L-glutamine (2 mM), penicillin (100 units/mL) and streptomycin (100 μg/mL) and then subcultured in DMEM supplemented with 10% FBS in a 37 ℃ incubator with 5% CO_2_.

### 4.2. Oligo-Fucoidan (LMF) and High Molecular Weight Fucoidan (HMF)

Oligo-Fucoidan (approximate molecular weight: 0.5–0.8 kDa, 92.1%) (Hi-Q Marine Biotech International Ltd., Taipei, Taiwan) is a glycolytic product of Fucoidan purified from the brown algae *Sargassum hemiphyllum* as described previously [21]. Briefly, freshly dried *Sargassum hemiphyllum* (100 g) was added to 5 liters of ddH_2_O and boiled at 100 °C for 30 min. The extracts were centrifuged and lyophilized under reduced pressure with 4 volumes of 95% ethanol for 24 h at 4 °C. Afterward, Fucoidan (5 g) was suspended in 125 mL of distilled water at 55 °C and stirred at 700 rpm, followed by digestion with a glycolytic enzyme at a concentration of 1 mg/g Fucoidan for 6 h and centrifugation at 10,000× *g* for 20 min at 4 °C. The supernatants were passed through a 30-kDa cut-off membrane and then a 1-kDa cut-off membrane (ProStream™ PP, TangenX Technology Co., Boston, MA, USA). Molecular weights were determined by injecting the samples into a high-performance size exclusion chromatograph (HPSEC) using an Ultrahydrogel 500 column (7.8 × 300 mm) (Waters, Milford, MA, USA) as described previously [58]. Purified Oligo-Fucoidan (purity ≥ 98%), which mainly contained L-fucose (210.9 ± 3.3 μmol/g) and sulfate ester (38.9 ± 0.4% (w/w)), was dissolved in PBS at a stock concentration of 20 mg/mL at 65 °C, filter-sterilized through a 0.22-μm MF-Minipore membrane filter (EMD Millipore, Darmstadt, Germany) and stored at −20 °C.

High molecular weight fucoidan (HMF) was obtained from Sigma-Aldrich (St. Louis, MI, USA).

### 4.3. Antibodies (Abs)

Abs against phospho-EGFR (Tyr1068), EGFR, GPX, phospho-p53 (Ser15), Cyclin B1, phospho-Histone H2A.X (Ser139), phospho-p38 MAPK (Thr180/Tyr182), p38 MAPK, iNOS, Arginase-1, TNF-α, IL-10, cleaved PARP (Asp214), PARP, caspase-3, active caspase-3 (Asp175), GPX and catalase were obtained from Cell Signaling (Danvers, MA). Abs recognizing p53 (EMD Millipore, Darmstadt, Germany), MnSOD (Enzo, Farmingdale, NY, USA), p21 (Santa Cruz, Dallas, Texas), p47^phox^ (Ab-345) (SAB, College Park, MD, USA), phospho-p47^phox^ (Ser345) (Elabscience, Houston, TX, USA), F4/80 and YY1 (GeneTex, Inc, Irvine, CA, USA), CD80 and CD163 (Abcam, Cambridge, MA, USA) were purchased as indicated. The SDS-PAGE and immunoblotting analysis was conducted as described previously [59].

### 4.4. Activation of Macrophage Polarization

THP-1 monocytes were induced by 100 μM phorbol 12-myristate 13-acetate (PMA) (Sigma-Aldrich) for 48 h to stimulated M0 macrophage differentiation. The M0 macrophages were induced to polarize into M1 macrophages by incubation with lipopolysaccharide (LPS) (100 pg/mL) (Cell Signaling) or into M2 macrophages by IL-4 (20 ng/mL) (Cell Signaling) for another 24 h. Macrophage polarity was confirmed by the mRNA expression levels of F4/80, CD68, CD80, CD86, CD163, CD206, TGF-β, Arginase-1 and IL-1β using quantitative RT-PCR or analyzed the amounts of F4/80, iNOS, p-p38 (Thr180/Tyr182), p38, CD80, TNF-α, CD163, Arginase-1 and IL-10 protein.

### 4.5. Mitochondrial Superoxide and Intracellular ROS Levels

HCT116 cancer cells were treated with DPI (1 μM) and/or Oligo-Fucoidan (400 μg/mL) and/or cisplatin (15 μM) (Sigma-Aldrich) for 48 h. The treated cancer cells were incubated with 5 μM MitoSOX^TM^ reagent (Molecular Probes, Eugene, OR, USA) for 10 min at 37 °C and the degree of MitoSOX Red oxidation was measured. The emitted MitoSOX fluorescence was quantified by a FACSCalibur flow cytometer (Becton-Dickinson, Franklin Lakes, NJ, USA) in the FL2 emission channel with an excitation wavelength of 543 nm. 

HCT116 cancer cells and THP-1 monocytes were treated with different doses of DPI for 24 h or 48 h and then incubated with 25 μM 2′,7′-dichlorodihydrofluorescein diacetate (DCF-DA) (Sigma-Aldrich) for 15 min at 37 °C. After DCF-DA was oxidized into fluorescent 2’,7’-dichlorofluorescein (DCF), the fluorescence signal emitted by DCF was measured by a FACSCalibur flow cytometer in the FL1 emission channel at a 488-nm excitation wavelength to detect intracellular ROS level.

Oligo-Fucoidan (400 g/mL), DPI (1 M) and NAC (1 M) were treated THP-1 monocytes or the M2 macrophages for 48 h to compare their effects on cellular ROS level by the flow cytometry analysis, as described above.

To exam the ROS signaling cascade, HCT116 cancer cells were pretreated with NAC (1 mM), MitoQ (1 M) and DPI (2 M) for 24 h followed by cotreatment of cisplatin (15 M) and Oligo-Fucoidan (400 g/mL) for 24 h. DPI and NAC were from Sigma-Aldrich. MitoQ was from Cayman Chemical (Ann Arbor, Michigan, USA).

### 4.6. Cell Cycle Analysis

HCT116 cancer cells were treated with Oligo-Fucoidan (400 μg/mL) and/or cisplatin (15 μM) for 48 h, fixed with 70% ethanol at −20 °C for 1 h and incubated with 0.1% (v/v) Triton X-100 (Sigma-Aldrich), 5 μg/mL DNase-free RNase A (Sigma-Aldrich) and 10 μg/mL propidium iodide (PI) (Thermo Fisher Scientific) in PBS in the dark for 20 min. PI fluorescence was analyzed by a BD FACSCalibur flow cytometer (Becton-Dickinson, San Jose, CA) in the FL2 emission channel at a 543-nm excitation wavelength.

### 4.7. Cytotoxicity Effect and Cell Viability Assay

HCT116 cancer cells and the primary C6P2-L1 colon cancer cells were treated with Oligo-Fucoidan (400 μg/mL) and/or cisplatin (15 μM) for 48 h, washed with ice cold PBS, resuspended in a 1× binding buffer (BD Biosciences, Franklin Lakes, NJ, USA) and then stained with 5 μL of FITC-conjugated Annexin V and 5 μL of PI in a total volume of 100 μL (1 × 10^5^ cells) for 15 min in the dark. Afterward, 400 μL of the 1× binding buffer was added to the samples, which were analyzed by a BD FACSCalibur flow cytometer in the FL1 and FL2 emission channels at excitation wavelengths of 488 nm and 543 nm, respectively. Early and late apoptosis events as well as necrosis effect were detected. 

THP-1 monocytes were treated by cisplatin (15 M) and/or Oligo-Fucoidan (400 g/mL) for 48 h. Cell viability were analyzed by WST-1 (4[3-(4-Iodophenyl)-2(4-nitrophenyl)-2H-5-tetrazolio]-1,3-benzene disulfonate) assay (Sigma-Aldrich) based on the cleavage of the tetrazolium salt to formazan by cellular mitochondrial dehydrogenase as described in the manufacturer’s instruction.

### 4.8. Analysis of CD80(+), CD86(+), CD163(+) and CD206(+) Macrophage Populations

THP-1 monocytes (1 × 10^6^) were treated with DPI (1 μM) for 48 h or treated with cisplatin (15 M) and/or Oligo-Fucoidan (400 g/mL) for 48 h. M0 macrophages (derived from THP-1 monocytes) were treated with cisplatin (15 M) and/or Oligo-Fucoidan (400 g/mL) for 48 h. M2 macrophages (derived from THP-1 monocytes) were treated with DPI (1 M), NAC (1 M), Oligo-Fucoidan (LMF) (400 g/mL) or HMF (400 g/mL) for 48 h. Raw264.7 macrophages (1 × 10^6^) were treated with cisplatin (15 μM) and/or Oligo-Fucoidan (400 μg/mL) for 48 h. The polarized macrophages were stained with FITC Mouse Anti-Human CD80 Ab (BD, Spark, MD) vs. FITC Mouse IgG1 κ Isotype Control (BD, Spark, MD), BB700 Mouse Anti-Human CD86 (BD, Spark, MD) vs. BB700 Mouse IgG1 κ Isotype Control (BD, Spark, MD), Alexa Fluor^®^ 647 Mouse Anti-Human CD163 (BD, Spark, MD) vs. Alexa Fluor^®^ 647 Mouse IgG1 κ Isotype Control (BD, Spark, MD, USA) or BB515 Mouse Anti-Human CD206 (BD, Spark, MD) vs. BB515 Mouse IgG1 κ Isotype Control (BD, Spark, MD) in a staining buffer (0.1% FBS in PBS) for 20 min at 4 °C, as indicated in each experiment. Fluorescence intensities were detected by a FACSCalibur flow cytometer in the FL1, FL3 and FL4 emission channels at excitation wavelengths of 488 nm and 633 nm, respectively.

The M2 macrophages (derived from Raw264.7 cells) treated with cisplatin (15 M) and/or Oligo-Fucoidan (400 g/mL) for 48 h were stained with PE Hamster Anti-Mouse CD80 (BD, Spark, MD) vs. PE Hamster IgG2, κ Isotype Control (BD, Spark, MD) and then analyzed by flow cytometry. Fluorescence intensity was detected by a FACSCalibur flow cytometer in the FL2 emission channels at excitation wavelengths of 488.

### 4.9. The Invasion Ability of M2 Macrophages and Cancer Cells

The polarized M2 macrophages (derived from THP-1 monocytes) were resuspended in serum-free medium, plated in the upper chamber of a Matrigel^®^ invasion transwell (24-well plate, 8-μm pore size) (Corning) and cocultured with DPI and/or Oligo-Fucoidan or cocultured with cisplatin and/or Oligo-Fucoidan in the bottom chamber for 24 h at 37 °C. The noninvasive M2 macrophages remaining on the upper membrane were removed by cotton swabs. The M2 macrophages penetrated through the Matrigel matrix were rinsed twice in PBS, fixed with 3.7% formaldehyde for 10 min, permeabilized with 100% methanol for 20 min, stained with 0.2% crystal violet for 15 min, photographed under an optical microscope and quantified by ImageJ software (version 1.52a, National Institute of Mental Health, Bethesda, Maryland).

HCT116 cancer cells (4 × 10^5^) suspended in serum-free medium were plated in the upper chamber of a Corning^®^ BioCoat^TM^ Matrigel^®^ invasion transwell (24-well plate, 8-μm pore size) (Corning, NY, USA) and cocultured with the DPI and/or Fucoidan-pretreated M2 macrophages in the bottom chamber for 24 h at 37 °C. The noninvasive cancer cells retained on the upper membrane were removed by cotton swabs. The invasive cancer cells through the Matrigel matrix were measured as described above. 

### 4.10. Expression Levels of Macrophage Markers Detected by Quantitative RT-PCR

The polarity of THP-1 monocytes and Raw264.7 macrophages as well as the derived M2 macrophages were studied after treatment with ROS inhibitors (NAC, DPI) (Sigma-Aldrich), DPI and/or Oligo-Fucoidan or cisplatin and/or Oligo-Fucoidan for 48 h. Moreover, HCT116 cancer cells (4 × 10^5^) or MDA-MB231 cancer cells (4 × 10^5^) were pretreated with cisplatin (15 μM) and/or Oligo-Fucoidan (400 μg/mL) for 48 h and the preconditioned cancer cells were rinsed and seeded in the upper chamber of Corning Falcon^®^ Cell Culture Inserts (Corning, NY, USA) and incubated with THP-1 monocytes (1 × 10^6^) placed in the bottom chamber for 48 h. Cellular RNAs of the polarized macrophages were isolated in a TRIzol solution (Thermo Fisher Scientific), treated with DNase I and transcribed into cDNA by SuperScript^TM^ II reverse transcriptase (Thermo Fisher Scientific) according to the manufacturer’s instructions. Quantitative RT-PCR was conducted using SYBR Green Master Mix, a cDNA template (100 ng) and the primers for F4/80 (forward: 5′-CAATGAGTGCCTCACCAGCA-3′; reverse: 5′-TGGGCAAGCTCTTGGATCTG-3′), CD80 (forward: 5′-GCAGGGAACATCACCATCCA-3′; reverse: 5′-TCACGTGGATAACACCTGAACA-3′), CD68 (forward: 5′-CAGGGAATGACTGTCCTCACA-3′; reverse: 5′-CTCTGTAACCGTGGGTGTCA-3′), CD86 (forward: 5′-GCTTTGCTTCTCTGCTGCTG-3′; reverse: 5′-GGCAGGTCTGCAGTCTCATT-3′), CD163 (forward: 5′-CCGGGAGATGAATTCTTGCCT-3′; reverse: 5′-AGACACAGAAATTAGTTCAGCAGCA-3′), CD206 (forward: 5′-CTGAATTGTACTGGTCTGTCCT-3′; reverse: 5′-GCTTAGATGTGGTGCTGTGG-3′), TGF-β forward: 5′-TTGACTTCCGCAAGGACCTC-3′; reverse: 5′-CTCCAAATGTAGGGGCAGGG-3′), IL-1β (forward: 5′-GAGCTCGCCAGTGAAATGAT-3′; reverse: 5′-GGCCATCAGCTTCAAAGAACAA-3′) and β-actin (forward: 5’-CACCAGGGCGTGATGGTGGG-3′; reverse: 5′-GATGCCTCTCTTGCTCTGGGC-3′) were used for analyzing of human cells. 

For murine cells, CD68 (forward: 5′-ACACTTCGGGCCATGTTTCT-3′; reverse: 5′-GGGGCTGGTAGGTTGATTGT-3′), CD80 (forward: 5’-ATGCTCACGTGTCAGAGGAC-3′; reverse: 5′-TGACAACGATGACGACGACT-3′), CD206 (forward: 5’-GTCAGAACAGACTGCGTGGA-3′; reverse: 5′-CAGCAGCAGTCTCGATGGAA-3′) and Arginase-I (forward: 5′-GGTCTGTGGGGAAAGCCAAT-3’; reverse: 5’-CAGTGTGAGCATCCACCCAA-3′).

The quantitative RT-PCR reaction was performed at 95°C for 15 min, followed by 40 cycles at 95 °C for 15 s and 60 °C for 1 min. The β-actin expression level was used as an internal control. Relative mRNA levels were calculated by the formula: ΔΔCT = ΔCt test sample−ΔCt control sample. Fold changes in gene expression were calculated using the 2^−^^ΔΔCT^ method.

### 4.11. Tumor Growth in Xenograft Mice and a Tumor Immunohistochemistry Study

Six- to eight-week-old BALB/c nude mice (BALB/cAnN.Cg-Foxn1^nu^/CrlNarl) were obtained from the National Laboratory Animal Center of Taiwan. The experiments were performed according to the Animal Use Protocol (NHRI-IACUC-105103-A) approved by the National Health Research Institutes (NHRI). Mice were injected subcutaneously with HCT116 cells (2 × 10^6^/100 μL of PBS) and randomly divided into 4 groups (PBS, Oligo-Fucoidan, cisplatin and combined treatment). Two weeks after inoculation, the tumors reached approximately 100 mm^3^ in size and the mice were intravenously (i.v.) injected with cisplatin (1 mg/kg) three times per week for 2 weeks and/or orally fed Oligo-Fucoidan (150 mg/kg) two times per week for 5 weeks. Control mice were administered PBS alone. Tumor volumes (Vs) were measured weekly and calculated using the formula: Vs = (length × width^2^)/2.

Tumor samples were processed for immunohistochemical (IHC) staining as described previously [60]. Abs against VEGF Receptor 2 (1:800, D5B1, Cell signaling) and CD163 (1:100, ab182422, Abcam) were diluted in PBS as indicated. IHC staining images were scanned by an automatic digital slide scanner (Pannoramic MIDI, 3DHISTECH Ltd., Budapest, Hungary) supported with Carl Zeiss objectives (40×). CD163 positive cells were randomly selected across each sample and quantified.

### 4.12. Disclosure

HLH serves on the Scientific Advisory Board of Hi-Q Marine Biotech International Ltd. The terms of this arrangement have been reviewed and approved by the National Health Research Institutes in accordance with its conflict of interest guidelines.

### 4.13. Statistics

A two-tailed unpaired Student’s t test was applied to compare the results of the control and experimental groups. The chi-square test was used to determine the statistical significance of differences in tumor burdens. A *p* value < 0.05 was considered statistically significant. 

## 5. Conclusion

We now confirm that Oligo-Fucoidan quenches intracellular ROS and mitochondrial superoxide which can directly or indirectly benefit M1-like macrophage polarization. Importantly, Oligo-Fucoidan suppresses the drawback of chemotherapy on M2 macrophage polarization *in vitro* and inhibits M2 macrophage infiltration *in vivo*. Our results first demonstrate that Oligo-Fucoidan supplementation sufficiently enhances chemo-sensitivity in aggressive cancer cells and renovates a healthy microenvironment that prevents tumor progression.

## Figures and Tables

**Figure 1 cancers-12-00421-f001:**
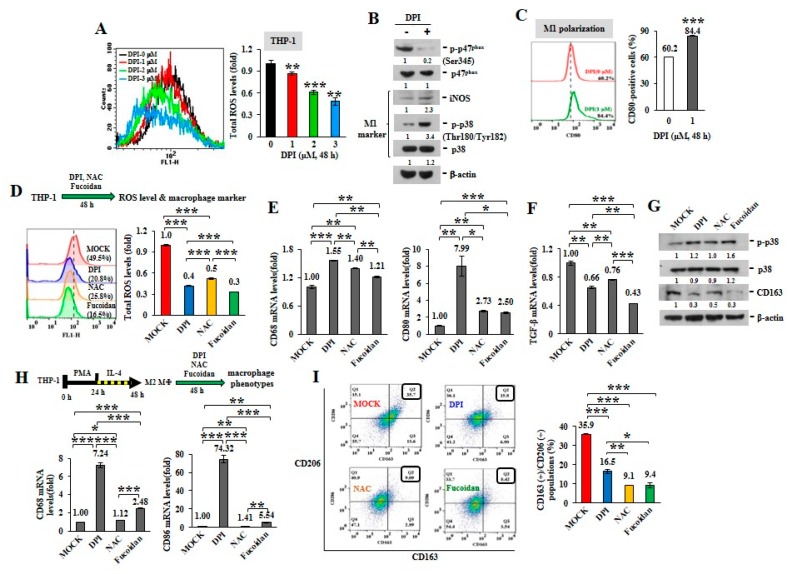
Reactive oxygen species (ROS) inhibitors and Oligo-Fucoidan regulate macrophage polarity. Unprimed THP-1 monocytes (**A**–**G**) and the M2 macrophages (**H**–**I**) were studied. Intracellular ROS levels were measured by 2′,7′-dichlorodihydrofluorescein diacetate (DCF-DA) assay after treatment with Diphenyleneiodonium (DPI) (0–3 M) for 48 h (**A**) The amounts of nicotinamide-adenine dinucleotide phosphate (NADPH) subunit p47^phox^, p-p47^phox^ (Ser345) and M1 marker (iNOS and p-p38 (Thr180/Tyr182)) were examined after DPI (1 M) treatment of monocytes for 48 h (**B**) The CD80(+) M1 macrophage populations were analyzed by flow cytometry (**C**) Total cellular ROS levels were assayed upon DPI (1 M), N-acetylcysteine (NAC) (1 M) and Oligo-Fucoidan (400 g/mL) treatment of monocytes for 48 h (**D**) The mRNA levels of M1 (CD68 and CD80) **(E)** and M2 (TGF-β) (**F**) markers were examined by quantitative reverse transcription polymerase chain reaction (RT-PCR) and the protein amounts of M1 (p-p38, p38) and M2 (CD163) marker were examined (**G**) M2 macrophages derived from THP-1 monocytes were stimulated with phorbol 12-myristate 13-acetate (PMA) (100 M) for 24 h and then IL-4 (20 ng/mL) for another 24 h. Upon DPI, NAC and Oligo-Fucoidan treatment of the M2 macrophages for 48 h, the mRNA levels of M1 marker (CD68 and CD86) were assayed by quantitative RT-PCR (**H**) and the CD163(+)/CD206(+) M2 macrophage populations were measured by flow cytometry **(I**) Data represent the mean ± standard deviation (SD) of three independent experiments (**A**,**C**,**D**,**E**,**F**,**H**,**I**) Student’s *t* test determined the statistical significance of pairwise comparisons; * *p* < 0.05; ** *p* < 0.01; and *** *p* < 0.001. The protein level was normalized with internal control (β-actin) and then compared with the level measured in MOCK treatment (**B**,**G**) The band intensity measured by ImageJ (http://rsb.info.nih.gov/ij/index.html) was indicated under each panel.

**Figure 2 cancers-12-00421-f002:**
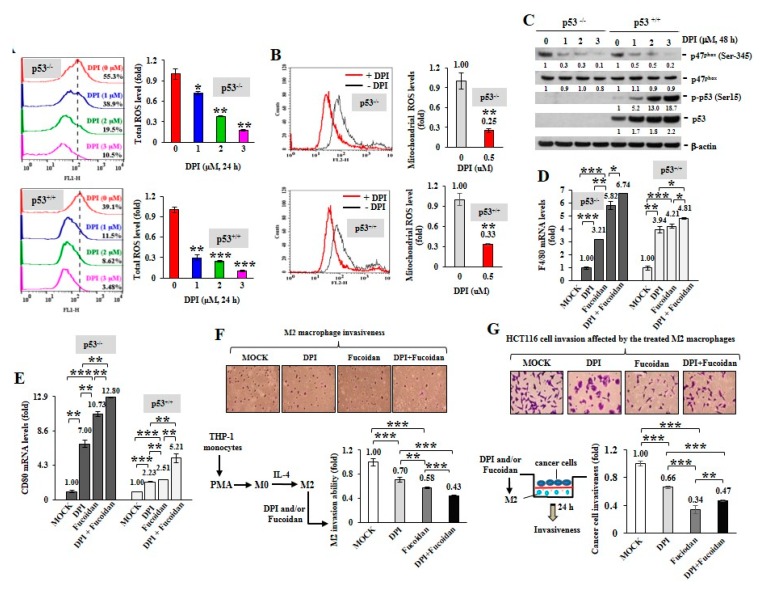
ROS inhibition in cancer cells induces M1 macrophage phenotypes. Total cellular ROS levels were examined upon DPI (0–3 M) treatment of HCT116 cancer cells (p53^+/+^ and p53^−/−^) for 24 h (**A**) Mitochondrial superoxide levels in HCT116 cells were measured after treatment with DPI (1 μM) for 1 h (**B**) NADPH subunit p-p47^phox^ (Ser345), p47^phox^, p-p53 (Ser15) and total p53 amounts were compared after treatment of HCT116 with DPI (0–3 M) for 48 h (**C**) Protein level was normalized to the β-actin protein level and then compared with the level of MOCK control. HCT116 cells were pretreated with DPI (1 μM) and/or Oligo-Fucoidan (400 g/mL) for 48 h, followed by incubation with THP-1 monocytes for 48 h in a Boyden chamber transwell. The surface markers of M0 (F4/80) (**D**) and M1 (CD80) (**E**) macrophages were assessed by quantitative RT-PCR. The M2 macrophage invasiveness was examined upon DPI and/or Oligo-Fucoidan treatment (**F**) The invasion ability of HCT116 cells was examined after coculture with the M2 macrophages those were pre-treated with DPI and/or Oligo-Fucoidan in the invasion transwell (**G**) Crystal violet staining of the invasive M2 macrophages and HCT116 cancer cells (**F**,**G**) The results are expressed as the mean ± SD (*n* = 3). Student’s *t* test determined the statistical significance of pairwise comparisons (* *p* < 0.05; ** *p* < 0.01; and *** *p* < 0.001).

**Figure 3 cancers-12-00421-f003:**
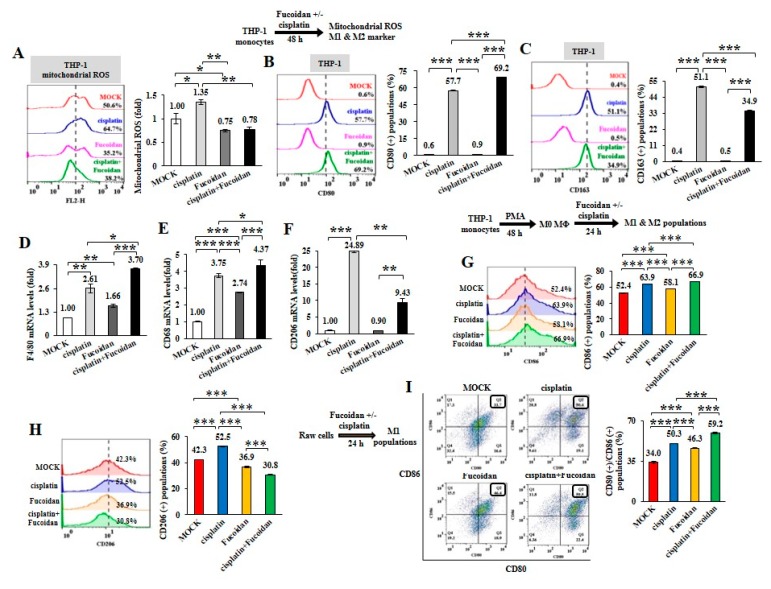
Cisplatin and/or Oligo-Fucoidan treatment directly affect macrophage polarity. THP-1 monocytes were treated with Oligo-Fucoidan (400 μg/mL) and/or cisplatin (15 μM) for 48 h. Mitochondrial superoxide levels were measured (**A**) The populations of CD80(+) M1 (**B**) and CD163(+) M2 (**C**) macrophages derived from the treated monocytes were examined by flow cytometry. The mRNA expression levels of M0 (F4/80) (**D**) M1 (CD68) (**E**) and M2 (CD206) (**F**) markers were evaluated by quantitative RT-PCR. The M0 macrophages derived from PMA-induced monocytes were studied upon cisplatin and/or Oligo-Fucoidan treatment for 48 h and the CD86(+) M1 (**G**) and CD206(+) M2 (**H**) populations were analyzed by flow cytometry. The results are expressed as the mean ± SD (*n* = 3). Student’s *t* test determined the statistical significance of pairwise comparisons (* *p* < 0.05; ** *p* < 0.01; and *** *p* < 0.001).

**Figure 4 cancers-12-00421-f004:**
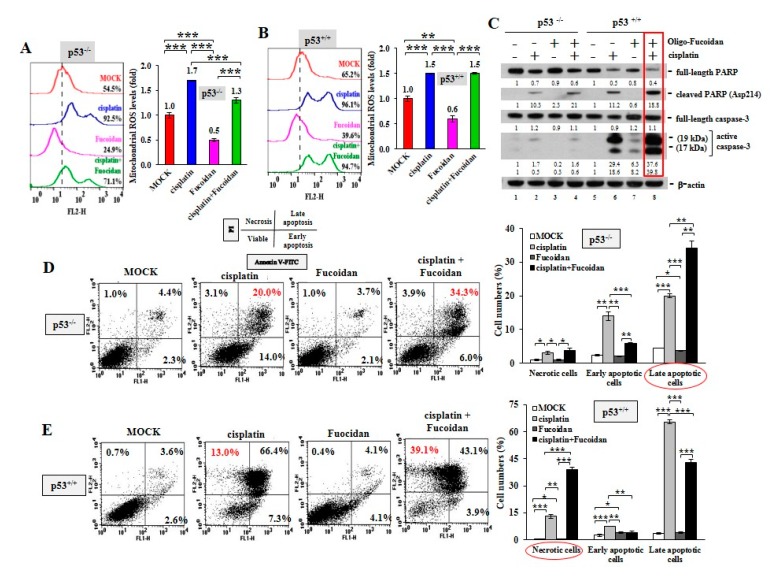
Oligo-Fucoidan and cisplatin together advance HCT116 cancer cell death. HCT116 cancer cells (p53^−/−^ and p53^+/+^) were studied upon Oligo-Fucoidan (400 μg/mL) and/or cisplatin (15 μM) administration for 48 h. (**A**,**B**) Mitochondrial superoxide levels were measured in HCT116 cells after receiving the indicated treatment. Histograms show the relative mitochondrial ROS levels in each group. (**C**) Apoptotic indicator (Poly (ADP-ribose) polymerase (PARP) cleavage and caspase-3 activation) in HCT116 cells were analyzed under the indicated conditions. The protein level was normalized to the β-actin level and compared with the level in MOCK control. (**D**,**E**) Cell death events (necrosis and apoptosis) occurring in the treated cells were detected by Fluorescein isothiocyanate (FITC)-conjugated Annexin V and propidium iodide (PI) staining and analyzed by flow cytometry. Histograms show the relative cell apoptotic and necrotic effects. The results are expressed as the mean ± SD (*n* = 3). Student’s *t* test determined the statistical significance of the indicated comparisons (* *p* < 0.05; ** *p* < 0.01; and *** *p* < 0.001).

**Figure 5 cancers-12-00421-f005:**
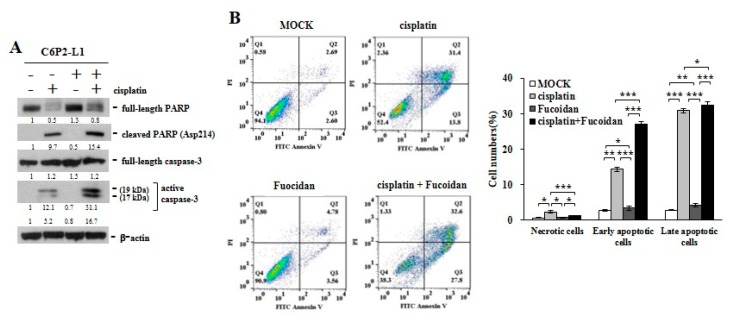
Oligo-Fucoidan and cisplatin cooperatively promote C6P2-L1 cancer cell death. Primary C6P2-L1 colon cancer cell were treated with cisplatin and/or Oligo-Fucoidan for 48 h. (**A**) The full-length PARP and cleaved PARP (Asp214) as well as the full-length capase-3 and active capase-3 were studied. (**B**) Cell death events were detected by FITC-conjugated Annexin V and PI staining, followed by flow cytometry analysis. Histograms show the percentages of necrotic and apoptotic cells in the indicated treatment.

**Figure 6 cancers-12-00421-f006:**
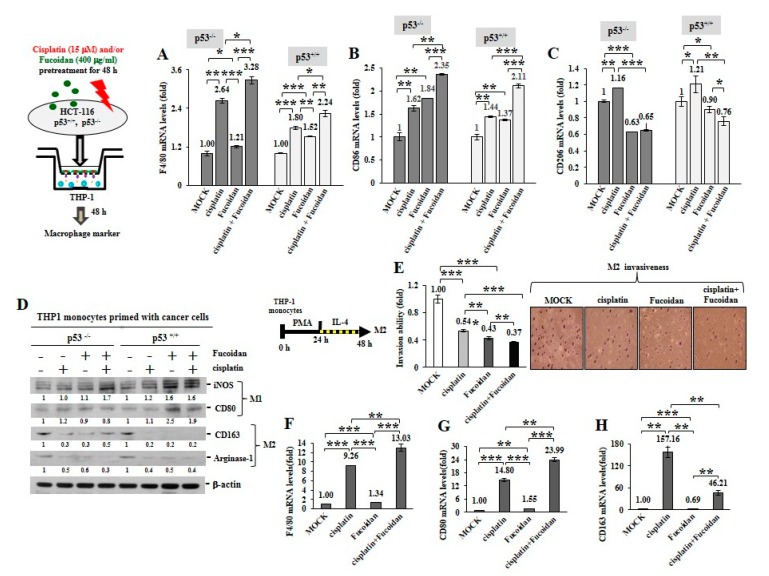
Oligo-Fucoidan and/or cisplatin-treated cancer cells influence macrophage plasticity. HCT116 cancer cells (p53^−/−^ and p53^+/+^) were pretreated with Oligo-Fucoidan (400 μg/mL) and/or cisplatin (15 μM) for 48 h and then cocultured with THP-1 monocytes for 48 h in a Boyden chamber transwell. The surface marker of M0 (F4/80) (**A**) M1 (CD86) (**B**) and M2 (CD206) (**C**) macrophages were evaluated by quantitative RT-PCR. Intracellular marker of M1 (iNOS and CD80) and M2 (CD163 and Arginase-1) macrophages were also examined (**D**) Protein level was normalized to the β-actin and compared with the level in cells receiving MOCK treatment. The M2 macrophages were derived from monocytes after stimulation with PMA (100 μM) for 24 h and then IL-4 (20 ng/mL) for another 24 h. The M2 macrophage invasiveness was analyzed upon Oligo-Fucoidan and/or cisplatin treatment for 24 h and then detected by crystal violet staining. (**E**) The M2 macrophages repolarized to M0 (F4/80) (**F**) and M1 (CD80) (**G**) phenotypes or maintained the M2 (CD163) polarity (**H**) were inspected in the indicated treatment. The results are expressed as the mean ± SD (*n* = 3). Student’s *t* test determined the statistical significance of pairwise comparisons (* *p* < 0.05; ** *p* < 0.01; and *** *p* < 0.001).

**Figure 7 cancers-12-00421-f007:**
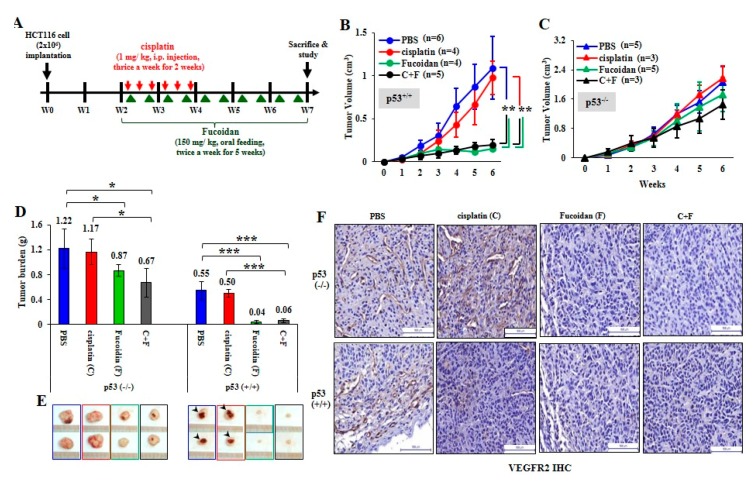
Oligo-Fucoidan prevents HCT116 tumor progression and angiogenesis. (**A**) HCT116 cancer cells (p53^−/−^ and p53^+/+^) (2 × 10^6^) were inoculated into nude mice and allowed tumor grow for 2 weeks, followed by treatment with cisplatin (1 mg/kg) 3 times per week for 2 weeks and/or Oligo-Fucoidan (150 mg/kg) twice per week for 5 weeks. Control experiments were treated with PBS alone. (**B**,**C**) The p53^−/−^ and p53^+/+^ tumor growth rates were assessed weekly. (**D**) Tumor burdens were measured and compared between different groups at week 7. (**E**) Representative photographs of p53^−/−^ tumors and p53^+/+^ tumors show the therapeutic outcomes. (**F**) Immunohistochemistry examined VEGFR2 (+) angiogenesis in the p53^−/−^ and p53^+/+^ tumors after the indicated treatment. The results are expressed as the mean ± SD. Student’s *t* test determined the statistical significance of pairwise comparisons (* *p* < 0.05; ** *p* < 0.01; and *** *p* < 0.001).

**Figure 8 cancers-12-00421-f008:**
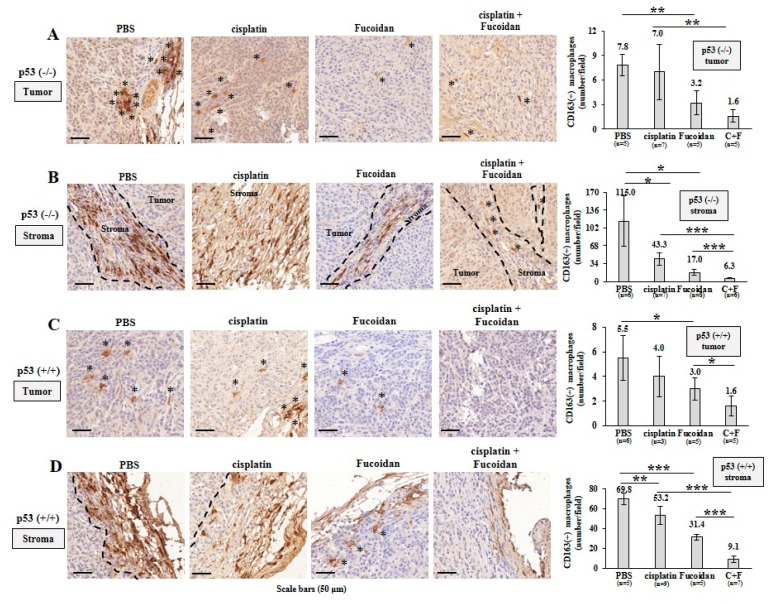
Oligo-Fucoidan alone or in combination with cisplatin inhibits M2 macrophage infiltration. Tumor-associated M2 macrophages were inspected by CD163 immunohistochemistry (IHC) after Oligo-Fucoidan and/or cisplatin treatment. Representative IHC images of CD163(+) M2 macrophages infiltrated into the tumoral (**A**) and stromal (**B**) compartments of p53^−/−^ tumors were examined. CD163(+) M2 macrophages in the tumoral (**C**) and stromal (**D**) regions of p53^+/+^ tumors were assessed. Histograms indicate the amounts of M2 macrophages in the designed therapies. The results are expressed as the mean ± SD. Student’s *t* test determined the statistical significance of pairwise comparisons (* *p* < 0.05; ** *p* < 0.01; and *** *p* < 0.001).

**Figure 9 cancers-12-00421-f009:**
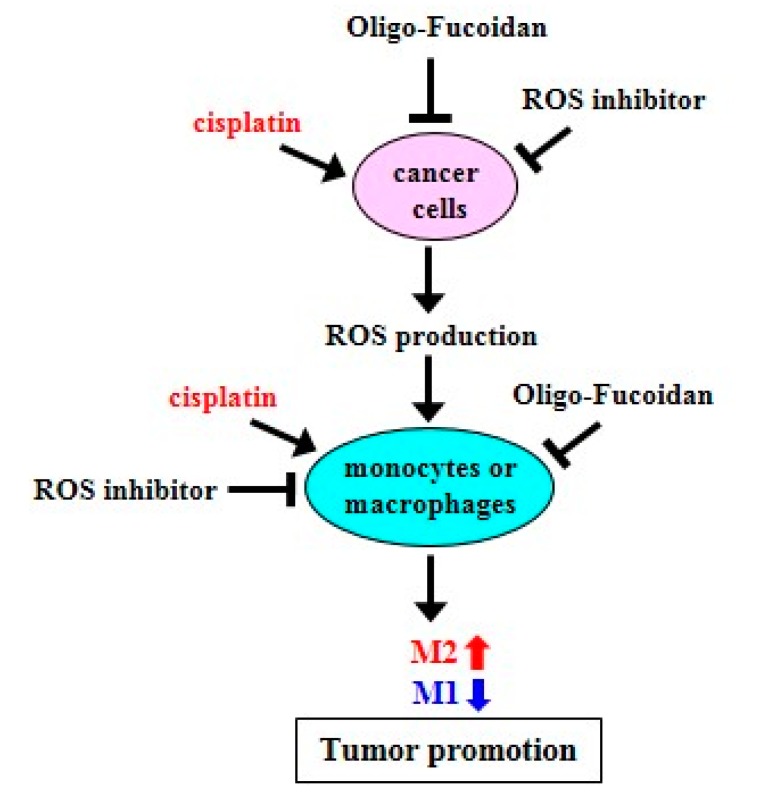
Schematic summary of Oligo-Fucoidan, ROS inhibitor and cisplatin effect on macrophage polarization and cancer treatment. Cisplatin induces ROS production but ROS inhibitor and Oligo-Fucoidan suppress ROS generation in aggressive cancer cells, which indirectly change macrophage polarity. Direct inhibition of ROS generation in monocytes or M2 macrophages by ROS inhibitor and Oligo-Fucoidan could synergistically promote M1 phenotype and prevent M2 polarity. Thereby, an antioxidant mechanism potentially establishes the tumor suppressive microenvironment(s) that prevent tumor progression.

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
