# Peer review of "Oligo-Fucoidan Prevents M2 Macrophage Differentiation and HCT116 Tumor Progression"

_cancers, 2020, doi:10.3390/cancers12020421_

Round 1
Reviewer 1 Report
The authors have sufficiently addressed my various comments and suggestions.
Reviewer 2 Report
Seemes to be acceptable.
This manuscript is a resubmission of an earlier submission. The following is a list of the peer review reports and author responses from that submission.
Round 1
Reviewer 1 Report
The authors showed anti-tumor effect of oligo-fucoidan was mediated by M1-polarization of macrophages. The presented data related to macrophage differentiation is not enough for support their hypothesis.
Comments
#1: It is well known that THP-1 cells do not express CD163 protein. Please check CD163 expression by western blot analysis. Negative control using isotype-matched IgG was not presented in all FACS data.
#2: Toxicity of cisplatin to THP-1 cells might be seen.
#3: Other M1/M2-related markers such as IL-10, IL-12, arginase1 should be tested. CD86 is a useful marker for M1/M2 polarization??
Reviewer 2 Report
Manuscript Summary
This manuscript from Chen et al., describes the use of oligo-fucoidan, isolated from brown seaweed, for the suppression of M2 macrophage polarization and diminished cancer cell viability as well as tumor burden in colorectal cancer as a cisplatin combination therapy. This compound shows significant antioxidative properties and the authors have nicely shown its capacity to slow or abolish tumor growth in vivo; particularly in p53 WT cells and tumor xenografts. This manuscript has many strengths, although a primary weakness is these studies are only performed in one cancer cell line and one in vitro monocyte/macrophage line, and some other items that warrant the authors attention are listed below. Regardless, these data are compelling and novel. Following the authors addressing the concerns below, the revised manuscript will be considered for publication in Cancers.
Overall Major Comments
Negative controls for flow cytometry histograms should be displayed for each surface marker and corresponding cell line and appended as an additional supplementary figure. Would be interesting to observe how Oligo-Fucoidan administration compares to a well described antioxidant, such as n-acetyl cysteine, or an ROS-scavenging approach e.g. catalase treatment. Inclusion of a side-by-side comparison in some of the more impactful experiments would significantly strengthen these findings. Some statement(s) made by the authors are a bit contradicting. Figures 1E, 3A, 4A/B, and lines 206-213, all discuss the antioxidative capacity of oligo-fuc administration, yet, line 228 describes how cisplatin-fuc treatment consistently amplified ROS signaling (i.e. sup fig 5). It is unclear how an antioxidant such as oligo-fuc would increase ROS signaling. The authors should attempt to clarify these statements. In parallel with item #3, the suggestion that cisplatin-fuc treatment promotes ROS signaling (sup fig. 5) needs additional controls to support these findings. These data show increases of overall EGFR levels (pY1068 vs total ratios are roughly the same in p53 KO (0.44) and p53 WT (0.6)), increases in levels of YY1, increases of MnSOD, and slight changes in GPX levels. Indeed, the authors have convincingly shown that cisplatin increases MtROS, and the increases in MnSOD are likely in response to the additional redox stress. The observed decreases in GPX levels (GPX-1 is assumed since not directly specified) are potentially an adaptation to utilize one of the other GPXs or the other H2O2 detoxification systems i.e. mitochondrial catalase, fenton rxns. While, EGFR and YY1 activity have been characterized to be regulated by direct cysteine modification via oxidants (EGFR - Truong and Carroll, Cell Chemical Biology 2016; YY1 – Hongo and Bonavida, Biochem Biophys Res Commun 2005), examination of EGFR/YY1 activation in p53 wt vs ko should be performed in combination with additional controls to limit redox-dependent modifications (i.e. NEM, NAC, catalase, DPI, others). The authors could perform the additional experiments needed to support these claims or adjust the text appropriately to reflect the data presented. In the discussion, the authors should address that these studies were performed in one cancer cell line and one monocyte line. Future studies should examine these approaches in additional colorectal cancer cell lines, PDXs, and primary human monocytes/macrophages to fully validate these findings.
Overall Minor Comments
Figure 5; p-EGFR is misspelled as p-EGFG; also in the figure p-EGFR (Try1068) should be Tyr1068 Line 343/344 is incorrectly stated. Fig 2F displays that M2 macrophages enhances cancer cell invasion, not inhibited. Line 355 – DPI is a NOX/DUOX inhibitor, which prevents the family of enzymes from generating ROS. DPI itself is not an ROS scavenger, like catalase or n-acetylcysteine.
Figure 1 Major Comments
Figure 1B is confusing or possibly mislabeled. The text cites that “The levels of NADPH subunit p47phox, p-p47phox and M1 markers were examined after DPI treatment and compared with those measured in MOCK treatment”. At first glance, it appears that THP cells were treated with only with mock in the first lane, subsequently measuring p-p47phox protein levels; and all the remaining lanes were treated with varying concentrations of DPI. I believe the authors are attempting to provide densitometric measurements of each lane. If the latter is the case, these data should be provided as a bar graph as an average of multiple experiments adjoined to the western blot images for clarity. At a minimum, this should be explained in the figure legend such not to confuse the reader. Potentially using the -/+ for DPI treatment nomenclature, akin to Fig. 1G. Would be of interest to examine NADPH oxidase expression levels pre/post LMF/HMF treatments to see if LMF/HMF are solely acting as antioxidants or downregulating NOX expression levels. Also, does LMF or HMF administration deplete overall oxidative stress? Only data for mitochondrial ROS is shown and/or described.
Figure 1 Minor Comments
Text font in the figure legend is at times inconsistent and missing symbols (i.e. μ) Unclear if DPI, LMF, HMF treatments were performed on unprimed THP-1 monocytes or if the treatments occurred post M0 differentiation with PMA. For future studies, alternative method to ID ROS other than DCFDA should be utilized. Many oxidative stress sensors and probes have intrinsic caveats and should be supported with alternative approaches to remove the potential for artifactual results.
Figure 2 Major Comments
Does pretreatment of M2 macrophages with DPI/Fuc suppress M2-induced cancer cell invasion? Would be an interesting and important continuation of Fig 2G.
Figure 2 Minor Comments
The authors should include a supplemental figure or edit the primary figure (Fig. 2F/G) displaying representative crystal-violet stained images that support the quantified invasion data. The brightfield/phase images shown in Fig 2F/G are difficult to quickly interpret and gather a conclusion.
Figure 3 Major Comments
See overall comment #2
Figure 3 Minor Comments
The exogenous addition of cisplatin and/or Fuc. to THP-1 monocytes and the effects of M1 vs M2 markers is clear and well described. It would be interesting to understand if these paradigms hold if cisplatin/Fuc. treatments are given post PMA differentiation of THP1 monocytes to M0 macrophages. This would potentially help address if locally administered (rather that given orally e.g. Fig. 6) fuc. could help promote M1 polarization once an unpolarized/unchallenged monocyte or macrophage enters the tumor microenvironment.
Figure 4 Major Comments
The authors quite nicely display that combination treatment of cis/fuc increases late apoptosis and necrosis in the two different p53 cell models with strong and compelling data, although, the mechanism of this remains unclear. Indeed, fuc treatment displays a strong antioxidative capacity via lowering levels of mitoROS as seen in fig 4A/B but correlating this to increasing apoptotic/necrotic events is a bit overstated; especially such that others have shown increasing levels of mitoROS induced by cisplatin contributes to its mechanism of action. It seems as if oligo-fucoidan harbors antioxidative properties, but this may not be the primary mechanism by which it enhances apoptosis/necrosis. The authors should address this in the results and discussion sections.
Figure 5 Minor Comments
Same remark as Fig. 2 minor comment #1 regarding fig 5E.
Figure 6 Minor Comments
The statement regarding decreased angiogenesis could be further supported with the use of alternative angiogenic markers i.e. VEGFR, LYVE-1. While not critical for the authors to address, it would help support this claim.
Figure 7 Minor Comments
Whole slide H&E images (1-2X magnification) of the tumor/stroma tissue sections should be included in the supplemental such that the reader may more effectively view the tumor margins. The Y axis label of the quantified IHC data should be changed to CD163+ macrophages. M2 macrophages should be defined via the identification of several molecular markers i.e. CD206+/CD163+/ARG-1+, etc.